# Cultivated Land Fragmentation and Its Influencing Factors Detection: A Case Study in Huaihe River Basin, China

**DOI:** 10.3390/ijerph19010138

**Published:** 2021-12-23

**Authors:** Jiale Liang, Sipei Pan, Wanxu Chen, Jiangfeng Li, Ting Zhou

**Affiliations:** 1School of Public Administration, China University of Geosciences, Wuhan 430074, China; LJL0715@cug.edu.cn (J.L.); pamsp@cug.edu.cn (S.P.); 2School of Geography and Information Engineering, China University of Geosciences, Wuhan 430074, China; 3Research Center for Spatial Planning and Human-Environmental System Simulation, China University of Geosciences, Wuhan 430078, China; 4State Key Laboratory of Earth Surface Processes and Resource Ecology, Beijing Normal University, Beijing 100875, China; 5School of Geographical Sciences, Fujian Normal University, Fuzhou 350007, China; zting1624@163.com

**Keywords:** cultivated land fragmentation, landscape pattern index, spatial autocorrelation, geographic detector, Huaihe River Basin, China

## Abstract

The booming population and accelerating urbanization in the Huaihe River Basin have sped up the land use transformation and the cultivated land fragmentation (CLF), seriously impeded the advancement of agricultural modernization, and threatened regional stability and national food security as well. The analysis of CLF degree and its spatiotemporal distribution characteristics, along with the influencing factors in the Huaihe River Basin, is of great significance for promoting the intensive and efficient utilization of cultivated land resources and maintaining food security. Previous studies lack the measurement and cause analysis of CLF in Huaihe River Basin. To bridge the gap, this study introduces Fragstats4.2 and ArcGIS10.3 to analyze the spatiotemporal characteristics of CLF in county units in the Huaihe River Basin from 2000 to 2018 through the Lorentz curve, entropy method, and spatial auto-correlation method while the causes of the spatiotemporal differentiation of CLF in the basin were explored with the help of a geographic detector. The results show that the spatial distribution of cultivated land in the Huaihe River Basin is relatively balanced, and the Gini coefficients of cultivated land from 2000 to 2018 were 0.105, 0.108, and 0.113, respectively. More than 56% of the counties in the basin have a location entropy greater than 1. the percentage of landscape, area-weighted mean patch area, patch cohesion index, and aggregation index decrease year by year while the patch density and splitting index show an upward trend. The landscape pattern of cultivated land is highly complex, and the overall fragmentation degree is increasing. The county distribution pattern of the CLF degree with random and agglomeration is generally stable. The spatiotemporal differentiation of CLF in the Huaihe River Basin is affected by multiple factors, among which the influences of the normalized difference vegetation index, per capita cultivated land area, and intensity of human activity obviously stronger than other factors, and the contribution rate of the factors reached more than 0.4. The interaction effect among the factors is stronger than that of single factor, with dual-factor enhancement and nonlinear enhancement dominating. The results of this study have important implications for optimizing the agricultural structure in the Huaihe River Basin and alleviating the CLF in important grain production areas.

## 1. Introduction

Serving as the fundamental resource for human survival and society development [1], cultivated land performs multiple functions, such as production and living, and plays an important role in maintaining ecological security and food security and promoting social and economic stability [2,3]. Since the 21st century, the disordered expansion of urban construction land [4,5], and the intensification of man-land contradiction brought about by the rapid development of social economy have forced land use changes and the intensification of cultivated land fragmentation (CLF) [6]. Cultivated land resources in China are facing serious threats, which, to a certain extent, hinder the development process of agricultural modernization and large-scale development [7], reduce the agricultural production efficiency [8], and threaten national food security and social stability. As an important grain production base in China, the Huaihe River Basin covers an area of only 2.9% of China’s land area, but its cultivated land area accounts for 12% of the country’s total [9]. In recent years, the degree of CLF in the basin has been deepened, and the food security has been threatened. Therefore, scientific measurement of the level and causes of CLF in the Huaihe River Basin has become a key to improve the efficiency of cultivated land use and to ensure regional food security.

CLF refers to the difficulty of concentrated and contiguous operation of cultivated land under the interference of human or natural factors, showing the state of interpolation, and scattered and disorderly utilization [1,2]. Corresponding to the scale operation of cultivated land, CLF not only features Chinese traditional agricultural production, but also exhibits one of the most distinctive characteristics of agricultural landscapes globally [10]. CLF can lead to an increase in the area of field cans and ditches, which directly results in the loss and waste of cultivated land resources [11]. A survey showed that cultivated land wasted by fragmentation in China accounts for about 19% of the net cultivated land areas [12], resulting in serious land waste. Simultaneously, some researchers have observed the increase of food production costs [13], and a reduction of productivity [14,15,16] and grain production as fragmentation increases. Moreover, ecological consequences invoked by fragmentation also include biodiversity loss, declined agricultural efficiency, and local micro-climate change, which will undoubtedly lessen the agricultural ecosystem provisioning service [17], further holding up China’s agricultural modernization. Currently, research touching on CLF has achieved fruitful results owing to the hard work of numerous scholars [18,19,20]. Current research on CLF mainly focuses on three aspects: the evaluation of CLF, the causes of the spatial differentiation of CLF, and the impact of CLF.

Pieces of evidence from previous literature have proved that landscape metrics can effectively externalize agricultural fragmentation [21], and thus have been widely acknowledged and utilized by academia [22]. Currently, related research on the evaluation of CLF most adopt the landscape pattern index to characterize the degree of CLF. However, the processing of the landscape pattern index differs. Most scholars prefer methods, such as the moving window [23], principal component analysis [17], and multiple linear regression [24], to comprehensively deal with the landscape pattern index. Nevertheless, in previous studies on CLF evaluation, the multi-collinearity among them was seldom considered in the selection of the landscape pattern index, which had a certain impact on the accuracy of the evaluation results. In the study of the causes of CLF, the impacts of social and economic factors, such as the urbanization rate, population density, land use degree, road traffic, and land property rights, and natural factors, such as altitude, slope, annual precipitation, and water network density, on CLF were mainly explored. Scholars have used different research methods and focused on different directions. Most of them used geographically weighted regression models to identify the influencing factors of CLF, focusing on both natural and socioeconomic factors [24,25,26,27,28,29]. According to the results of scholars’ studies, the influencing factors of CLF in different study areas are various, with some scholars considering soil quality diversity as the most important influencing factor of CLF [25], while others believe that socioeconomic factors play a more important role [26]. However, in the analysis of the causes of CLF, only the macro-qualitative analysis of the influencing factors of CLF was carried out, while the influence of each factor was not specifically quantified. Moreover, only the individual effects of each factor have been analyzed, but the interaction between the influencing factors has not been explored yet.

A review of previous studies found that a wide range of research areas of CLF tend to use qualitative analysis, and the quantitative analysis research mostly takes provinces or districts as the research object, but there is a lack of CLF measurement and influencing factors detection research on the watershed scale. It is necessary to analyze important grain-producing areas of the CLF mechanism from the perspective of the basin, which will provide a reference for ensuring food security and sustainable utilization of cultivated land resources. This study focuses on the following three aspects:(1)The degree of CLF and the balance of the spatiotemporal distribution of cultivated land resources in the Huaihe River Basin from 2000 to 2018 were measured by the entropy method and Lorentz curve method.(2)The spatial auto-correlation method was introduced to detect the spatial clustering characteristics of high and low values of CLF change in county units.(3)In cooperation with the geographical detector, the influencing factors of spatiotemporal differentiation of CLF were explored.

The purpose of this study is to improve the agricultural modernization level and ensure food security of the Huaihe River Basin, and to provide a scientific reference for evaluating the CLF from the perspective of the basin and studying the driving mechanism of the CLF in important grain production areas.

## 2. Materials and Methods

### 2.1. Study Area

Covering a total watershed area of 270,000 km^2^, the Huaihe River Basin is located in eastern China, between the Yangtze River and the Yellow River (Figure 1). It is divided into the Huaihe River and the Yishusi River with the abandoned Yellow River as the boundary. The Huaihe River enjoys a pleasant climate, with the south a subtropical zone and the north a warm temperate zone. The overall terrain of the basin is flat, dominated by plain, with extensive cultivated land. The total cultivated land area is 12.7 million hectares, accounting for 11.7% of China’s total cultivated land area [30]. The grain output reaches 1/6 of the total grain output in China, making it an important grain production base in China. In recent years, with rapid development of the social economy and the acceleration of urbanization and industrialization in the Huaihe River Basin, the problem of CLF has become increasingly severe. The increase of CLF not only reduces the technical efficiency of cultivated land utilization [31], but also hinders the improvement of the scale efficiency of agricultural production and the process of agricultural modernization [14], which exerts a certain impact on the utilization of cultivated land resources and food production capacity in the Huaihe River Basin [32], threatening the food security and regional stability of the basin. According to the study of Zhou et al. [33], this study identified the Huaihe River Basin as the four provinces of Anhui, Henan, Shandong, and Jiangsu. A total of 218 counties in the four provinces were selected as the research objects, and the degree of CLF and its causes in the basin were analyzed from the county unit scale to provide ideas for improving the grain production efficiency in the Huaihe River Basin and solidifying its status as a grain production base.

### 2.2. Data Sources

The remote sensing monitoring data of land use in 2000, 2010, and 2018, annual average precipitation, road network, rivers, and normalized difference vegetation index (NDVI) of each county used in this study are all from the Resource and Environment Science and Data Center, Chinese Academy of Sciences (http://www.resdc.cn/ (accessed on 10 May 2021)). Among them, land use data is generated through manual visual interpretation, with a spatial resolution of 1 km and a comprehensive accuracy of over 90%. According to the national land use classification system, the land use types were divided into seven first-level types: cultivated land, forest land, grassland, water area, urban and rural construction land, unused land, and wetland [34]. Altitude data were obtained using 90 m resolution ASTERG DEM data from Geospatial Data Cloud (http://www.gscloud.cn/ (accessed on 15 May 2021)). The population density data is from the WorldPop (https://www.worldpop.org/ (accessed on 11 July 2021)), with a resolution of 100 m.

### 2.3. Method

To explore the CLF and its influencing factors in the Huaihe River Basin, this study first measured the spatiotemporal distribution characteristics of cultivated land in the Huaihe River Basin with the Lorentz curve, and then the degree of cultivated land fragmentation based on the entropy weight method was measured, and the spatial agglomeration characteristics of CLF with the spatial autocorrelation tool were analyzed. Eventually, the primary influencing factors of CLF in Huaihe River Basin were explored by the geographical detector (Figure 2).

#### 2.3.1. Lorentz Curve

The Lorentz curve, proposed by Lorentz, an American statistician, is widely applied to reflect the fairness of income distribution in a country or region. In recent years, more and more studies have introduced the Lorentz curve into studies on cultivated land resource change, food security, land use structure analysis, and land use transition, etc. [35,36,37]. In this study, the Lorentz curve was introduced to identify the concentration degree of cultivated land spatial distribution in the Huaihe River Basin from 2000 to 2018, and on this basis, the difference degree of cultivated land distribution in different county units was further described quantitatively through the calculation of Gini coefficients, providing a basis for the measurement of CLF in the Basin. The calculation equation is as follows:(1)$$Q=\frac{P_{1}/P_{2}}{P_{3}/P_{4}},$$
where *Q* is the location entropy of cultivated land in the study area, also known as the specialization rate; *P*_1_ is the cultivated land area of a county in the study area; *P*_2_ is the total area of cultivated land in the study area; *P*_3_ is the total land area of a county; and *P*_4_ is the total land area of the study area. The *Q* value can reflect the balance degree of cultivated land distribution in the basin. When *Q* < 1, it indicates that the proportion of cultivated land area in the county is less than that of the total land area in the study area, namely, the specialization degree is low, and the county is at a disadvantage. Otherwise, when *Q* > 1, it indicates that cultivated land has a high degree of specialization and is a regional advantage.

According to Equation (1), the location entropy *Q* of cultivated land is calculated, and the *Q* value is sorted from small to large. The cumulative percentage of cultivated land area and land area in each county is calculated, and then the cumulative percentage of cultivated land area in each county is used as the horizontal coordinate and the cumulative percentage of cultivated land area is plotted on the ordinate to draw the Lorentz curve.

Gini coefficient is used to describe the uniformity of the spatial distribution of research objects in the research area and to quantify the Lorentz curve. The calculation equation is as follows:(2)$$G=\sum_{i=1}^{m-1}\left(P_{i}Q_{i+1}-P_{i+1}Q_{i}\right),$$
where *G* is the Gini coefficient; $P_{i}$ represents the cumulative percentage of a county’s land area in the total land area of the study area; and $Q_{i}$ represents the cumulative percentage of cultivated land area of a county in the total cultivated land area of the study area. With reference to related research [38], the larger *G* is, the more uneven the distribution of cultivated land in the study area is. When *G* < 0.2, the distribution is uniform; 0.2 < *G* < 0.3, the distribution is relatively uniform; 0.3 < *G* < 0.4, the distribution is basically reasonable; 0.4 < *G* < 0.6, the distribution difference is large; and *G* > 0.6, the distribution difference is over the top.

#### 2.3.2. Cultivated Land Fragmentation Index Selection

The degree of CLF is affected by different factors, such as the shape, size, and connectivity of cultivated land blocks, which cannot be simply described by a single dimension index. With reference to previous studies [1,24,39] and the actual situation of the Huaihe River Basin, 10 landscape indicators were selected from three aspects: size, edge-shape, and aggregation, to characterize the degree of CLF in the Huaihe River Basin. They are the percentage of landscape (PLAND), patch density (PD), edge density (ED), area-weighted mean patch area (AREA_AM), area-weighted mean shape index (SHAPE_AM), area-weighted mean patch fractal dimension (FRAC_AM), interspersion and juxtaposition index (IJI), patch cohesion index (COHESION), splitting index (SPLIT), and aggregation index (AI).

To reduce redundant indicators and improve the accuracy of the evaluation model, SPSS22 software (IBM, Armonk, NY, USA) was applied to test the multi-collinearity of the evaluation indicators to determine whether a variable should be excluded and the reciprocal of tolerance, namely the variance inflation factor (VIF), can be used for testing. When VIF > 10, it indicates that the multi-collinearity of this variable is very serious, which will affect the parameter estimation of the evaluation model and should be considered to remove this variable [40]. The multi-collinearity diagnosis results of various landscape metrics are shown in Table 1.

According to the test results above, the area-weighted mean shape index (SHAPE_AM) and area-weighted fractal dimension (FRAC_AM) with VIF > 10 are finally removed to maintain the accuracy of the evaluation results. The calculation equation and ecological significance of each landscape metric are shown in Table 2 [41,42].

#### 2.3.3. Measurement of Cultivated Land Fragmentation

To reflect the degree of CLF in the basin from 2000 to 2018 more intuitively, the entropy weight method is introduced to determine the weight of the selected landscape indicators on the degree of CLF in the basin. The specific steps are as follows [43]:(1)Data standardization

To avoid the inconsistency of the indicator units from affecting the calculation weight, the range method is adopted to standardize the data and the indicator data is converted to a range between 0 and 1. The larger the positive indicator value, the closer the evaluation target value is to the ideal value. The larger the negative indicator value, the more the evaluation target deviates from the ideal value.

Positive indicator:(3)$$y_{ij}^{\prime }=\frac{y_{ij}-min\left\{y_{ij}\right\}}{max\left\{y_{ij}\right\}-min\left\{y_{ij}\right\}}$$

Negative indicator:(4)$$y_{ij}^{\prime }=\frac{max\left\{y_{ij}\right\}-y_{ij}}{max\left\{y_{ij}\right\}-min\left\{y_{ij}\right\}}$$
where $y_{ij}^{\prime }$ represents the standardized value of indicator *j* in the year *i*; $y_{ij}$ represents the actual value of indicator *j* in the year *i*; and $max\left\{y_{ij}\right\}$ and $min\left\{y_{ij}\right\}$ represent the maximum and minimum values of the indicator *j*, respectively.

(2)Calculation of the standardized value *P_ij_* of indicator *j* in the year *i*:


(5)
Pij=yij′∑i=1nyij′


(3)Calculation of the entropy value *e_j_* of indicator *j*:

(6)$$e_{j}=-k\overset{n}{\underset{i=1}{\sum }}P_{ij}=-\frac{1}{\ln\;m}\overset{n}{\underset{i=1}{\sum }}P_{ij}\ln\;P_{ij}$$
where *k* represents the proportionality coefficient, *k* = 1/ln *m*, and *m* is the number of research samples.

(4)Calculation of weight *W_j_* of indicator *j*:


(7)
$$W_{j}=\left(1-e_{j}\right)/\overset{n}{\underset{j=1}{\sum }}\left(1-e_{j}\right)$$


(5)Calculation of composite scores:

(8)$$Z=\overset{m}{\underset{j=1}{\sum }}\left(W_{j}\cdot P_{ij}\right)$$
where Z is the composite score of the CLF index, $W_{j}$ is the weight coefficient of indicator *j*, and $P_{ij}$ is the standardized value of indicator *j*.

#### 2.3.4. Spatial Autocorrelation

Spatial autocorrelation is often used to detect the potential interdependence between geographic data within a region [44]. With the help of Geoda1.12, this study uses the local spatial auto-correlation method to detect the random mode, discrete mode, and clustering characteristics of the spatial distribution of CLF in the Huaihe River Basin, and performs visual analysis. According to the spatial location of county units and the change of the CLF degree in the Huaihe River Basin, Local Moran’s I statistic was introduced to measure the spatial autocorrelation of CLF degree change in the Huaihe River Basin from 2000 to 2018. The Local Moran’s *I* can be calculated using Equation (9):(9)$$Local\;Moran^{\prime }s\;I=\frac{\left(x_{i}-\bar{x}\right)}{S^{2}}\overset{m}{\underset{j=1}{\sum }}w_{ij}\left(x_{j}-\bar{x}\right)$$
(10)$$S^{2}=\frac{1}{m}\overset{m}{\underset{i=1}{\sum }}{\left(x_{i}-\bar{x}\right)}^{2}$$
where *m* is the number of county units; *x_i_* and *x_j_* are the measured values of spatial unit attributes, respectively; $\bar{x}$ is the mean value of the measured value; and $w_{ij}$ is the spatial weight matrix, and $S^{2}$ is its variance.

#### 2.3.5. Geographical Detector

Geographical detector is a statistical method to detect the spatial differentiation characteristics of geographical phenomena and reveal their influence [45], which has been widely applied in multiple disciplines (e.g., environmental science and resource utilization, regional economy, physical geography, tourism, agricultural basic science, etc.). It consists of factor detection, interactive detection, ecological detection, and risk detection. This study mainly uses factor detection and interactive detection methods to distinguish the influence of the driving factors of the spatial differentiation of CLF in the Huaihe River Basin, and reveals the correlation between the driving factors. The calculation is as follows:(11)$$q=1-\frac{1}{N\sigma^{2}}\overset{L}{\underset{i=1}{\sum }}N_{i}\sigma_{i}^{2}$$
where *q* is the influence of the driving factor, *q*∈[0,1]; *N* is the number of samples in the study area, and *i* is the partition (*i* = 1,2,…, *L*); and $\sigma^{2}$ and $\sigma_{i}^{2}$ are the variances of indicators in the study area and the variances of partition *i*, respectively. The size of *q* reflects the degree of spatial differentiation of the indicators. The larger the *q* value, the stronger the explanatory ability of each factor to the dependent variable, and vice versa.

Cultivated land is a complex of natural environment and socioeconomic conditions composed of topography, soil, climate, hydrology, vegetation, and human socioeconomic activities, which is affected by multi-dimensional factors. Considering previous studies [1,14,24] and data availability, this study selected the following indicators from two aspects of natural resource endowment and socioeconomic development as the influencing factors for the spatial distribution of CLF in the Huaihe River Basin. Among them, natural factors are the basis for the formation of the spatial pattern of cultivated land resources in the Huaihe River Basin. Altitude (X1) and slope (X2) directly affect the distribution and utilization of cultivated land. The distance to the river (X3), NDVI (X4), and average annual precipitation (X5) provide important conditions for the distribution and development and utilization of cultivated land resources. Socioeconomic factors mainly reflect the interference degree of different human activities on the cultivated land landscape pattern, including per capita cultivated land area (X6), intensity of human activities (X7), population density (X8), and distance from road network (X9). The selected influencing factor data is processed by the Arc Toolbox/Spatial Analyst Tools/Reclass and Arc Toolbox/Spatial Analyst Tools/Zonal/Zonal Statistics tools of ArcGIS10.3 (Esri, Redlands, CA, USA) for discretization and classification. The spatialization and quantification of the influencing factors of county units within the basin was realized, and the specific classification methods and descriptions are shown in Table 3.

## 3. Results

### 3.1. Spatiotemporal Distribution Characteristics of Cultivated Land

#### 3.1.1. General Pattern of the Spatiotemporal Distribution

According to the statistics of land use data in the study area, cultivated land, forest land, grass land, and unused land decreased from 2000 to 2018. The most obvious change was in grass land, with a 22.08% reduction. Construction land and water area continued to increase, with increases of 27.96% and 12.17%, respectively. Specifically, there were 183,751 km^2^ of cultivated land in the Huaihe River Basin in 2018, of which the dry land area was 134,591 km^2^, accounting for 73.25% of the total cultivated land area, and the paddy land area was 49,160 km^2^, accounting for 26.75% of the total cultivated land area. From the perspective of time scale, the cultivated land area decreased by 2363 km^2^ from 2000 to 2010, and 6524 km^2^ from 2010 to 2018, indicating that the cultivated land area in the Huaihe River Basin decreased year by year and the reduction amplitude increased. According to the spatial distribution of land use in the study area (Figure 3), the spatial distribution of cultivated land in the Huaihe River Basin is relatively balanced, and the spatial distribution boundary between dry land and paddy field is clear. Dry land is mainly distributed in the area north of the Huaihe River while paddy field is mostly distributed in the area south of the Huaihe River, and a small amount is also distributed in Xuzhou, Suqian, and other cities north of the Huaihe River.

#### 3.1.2. Balance of Spatiotemporal Distribution

To further explore the balance of the spatiotemporal distribution of cultivated land resources in the Huaihe River Basin during the study period, Equation (1) was used to calculate the location entropy *Q* of the cultivated area of each county in the basin, and then plot the cultivated land Lorentz curve in the Huaihe River Basin in 2000, 2010, and 2018, respectively (Figure 4), and the Gini coefficient of the cultivated land according to Equation (2) was calculated. The results showed that the Lorentz curve of cultivated land in the Huaihe River Basin was close to the absolute average line from 2000 to 2018, and the variation range was small, indicating that the spatial distribution of cultivated land resources in the basin was relatively scattered during the study period. The Gini coefficients of cultivated land in 2000, 2010, and 2018 were 0.105, 0.108, and 0.113, respectively, increasing year by year but less than 0.2, illustrating that the spatial distribution of cultivated land in the Huaihe River Basin was relatively uniform during the study period. According to the spatial distribution of the cultivated land locational entropy in the Huaihe River Basin from 2000 to 2018 (Figure 5), the locational entropy of cultivated land in more than 56% of the counties in the basin was greater than 1, indicating a high degree of specialization of the cultivated land in the basin. Simultaneously, the location entropy shows a coexistence of an increase and decrease, but the increasing trend dominated, indicating that the degree of cultivated land specialization in Huaihe River Basin increased. Specifically, the areas with a low location entropy of cultivated land were concentrated in Lu’an, Xinyang, and Zibo—Xuzhou—Huai’an, revealing that these areas are inferior areas of cultivated land with low specialization, which is possibly attributed to the rapid economic development and excessive occupation of cultivated land by urban expansion in some areas. From 2000 to 2018, the location entropy of some regions decreased significantly, most notably in Zhengzhou, Kaifeng, and Pingdingshan, indicating that the level of cultivated land agglomeration and specialization showed a conspicuous decrease. The location entropy of cultivated land in Zhumadian, Bozhou, Bengbu, and other cities increased significantly, and the agglomeration trend was constantly enhanced.

### 3.2. Spatiotemporal Distribution of Cultivated Land Fragmentation

#### 3.2.1. Spatiotemporal Distribution of the Cultivated Land Fragmentation Index

According to the measurement results of the CLF index in the Huaihe River Basin from 2000 to 2018 (Table 4), four negative indicators that characterize the CLF, namely PLAND, AREA_AM, COHESION, and AI, all show a decreasing trend year by year, which to some extent reflect that CLF increased in the Huaihe River Basin during the study period. Whereas PD and SPLIT of the positive indicators increased year by year, implying that the CLF in the Huaihe River Basin deteriorated. Among them, PLAND continued to decrease, and the decreasing amplitude grew, indicating that the composition of the cultivated land landscape in the basin increased and diversified. ED first increased and then decreased, indicating that the edge shape of the cultivated land landscape elements in the basin changed from irregular to regular, and the area of cultivated land patches also fluctuated significantly. COHESION decreased year by year, illustrating that the physical connection between cultivated land patches in the basin continued to decrease, and the phenomenon of cultivated land patch dispersion and fragmentation was severe. SPLIT kept on increasing and the increase rate gradually enlarged, demonstrating that cultivated land patches tended to be scattered, and the cultivated land landscape became more and more fragmented in the basin during the study period.

The spatial distribution of the cultivated land landscape indexes PLAND and COHESION in the Huaihe River Basin differed significantly and had clear boundaries during 2000–2018 (Figure 6 and Figure 7). The high value areas were concentrated in some counties of Henan and Anhui in the middle and west of the basin while the low value areas were only distributed in Zaozhuang, Xinyang, and Lu’an. During the study period, a large number of PLAND areas changed from high to low values, with significant changes in most counties in Huaibei, Zhengzhou, Heze, and other cities, which is closely related to the decreasing amount of cultivated land in these areas. The transition from a middle- and low-value area to a high-value area was rare, and the change was more obvious in some counties of Suzhou and Linyi, principally because of the remarkable achievement of land development and reclamation in these counties and the obvious increase of cultivated land patches. PD showed little difference in the spatial distribution (Figure 8). Low-value areas were widely distributed, and a small number of medium-value areas and high-value areas were scattered in some counties of Lu’an, Xinyang, Zibo, and other cities. During the study period, the spatial distribution did not change much. Only a very small number of counties in Zhengzhou, Huainan, Linyi, and other cities changed from low-value areas to medium- to high-value areas. The high-value areas of ED were concentrated in the northeast of the basin while the low-value areas were concentrated in Yancheng and Nantong in the east of the basin and some southern counties, such as Lu’an (Figure 9). The spatial distribution characteristics of AREA_AM and IJI were relatively similar (Figure 10 and Figure 11). During the study period, the high-value areas of these two types were predominantly distributed in the south of the Huaihe River, whereas the low-value areas were concentrated in the middle of the basin. Some counties, such as Xuchang, Nanyang, and Lu’an, changed from high-value areas of AREA_AM to low-value areas, and only a few counties like Yancheng changed from low-value areas to high-value areas. Some counties, such as Xinyang, Huainan, and Huaibei City, changed from high-value areas of IJI to low-value areas, and only a few counties, such as Linyi City, changed from low-value areas to high-value areas. Low-value areas and high-value areas of SPLIT showed wide spatial distribution differences and clear boundaries (Figure 12). Low-value areas were concentrated in the central and eastern counties of the basin, whereas high-value areas were scattered, with Zibo, Huai’an, and Lu’an in the majority. During the study period, Luoyang, Xinyang, and Huai’an changed from low-value areas to high-value areas with significant changes. The spatial distribution of AI was balanced, with only a small number of low-value areas distributed in Lu’an and Zibo (Figure 13), whereas high-value areas were only distributed in some counties in Yancheng and Nantong.

#### 3.2.2. Spatiotemporal Distribution Characteristics of Comprehensive Index of Cultivated Land Fragmentation

According to the entropy weight method, the spatiotemporal distribution of the comprehensive fragmentation index of cultivated land from 2000 to 2018 was obtained (Figure 14). Overall, the spatial heterogeneity and complexity of the cultivated land landscape pattern in the Huaihe River Basin was comparatively high while the spatiotemporal differences of fragmentation varied significantly. The degree of fragmentation was high in the north and south of the basin but low in the middle. The overall degree of CLF deepened, but in some regions, the degree of fragmentation exhibited both increasing and decreasing trends. In 2000, Zibo, Zaozhuang, and Lu’an were the most seriously fragmented areas, whereas Bozhou, Huaibei, and Yancheng were the least fragmented. In 2010, areas with severe CLF in the basin had not yet been improved, and the degree of fragmentation in Zhengzhou, Jining, and other cities and the Bengbu-Huai’an-Taizhou line was significantly deepened. In 2018, the overall fragmentation of the basin further deteriorated, and the fragmentation degree of counties in Zhengzhou, Xuzhou, Taizhou, and other cities continued to deepen, while only some counties in Zhoukou, Shangqiu, Linyi, and other cities showed a decrease in the fragmentation degree. Additionally, it is worth noting that from 2000 to 2018, except for Hefei and Lu’an, the degree of CLF deepened in other counties along the Huaihe River, and the degree of CLF changed dramatically in the area of Huainan, Huai’an, and Yancheng, with a relatively high degree of fragmentation.

### 3.3. Correlation Analysis of Spatial Distribution of Cultivated Land Fragmentation

To further explore the spatiotemporal distribution characteristics of the comprehensive index of CLF in the Huaihe River Basin from 2000 to 2018, Geoda 1.12 was applied in this study to conduct a univariate local spatial auto-correlation analysis of the comprehensive index of CLF in the basin during the study period, and a local indicators of spatial association (LISA) map was generated, namely, the aggregation distribution pattern of the comprehensive fragmentation index. In the LISA map, the comprehensive fragmentation index of the counties in the four quadrants of high-high, low-low, low-high, and high-low was significant at the level of 5%. To better display the comprehensive degree of fragmentation of each county in the basin, ArcGIS10.3 was used for visual analysis of changes in 2000, 2010, and 2018 (Figure 14). As shown in Figure 15, the distribution pattern of random and agglomeration counties in the Huaihe River Basin was generally stable during the study period. High-high areas represent the agglomeration areas of high value and high value, which were concentrated in some counties in Linyi, Zaozhuang, Pingdingshan, and other cities. The altitude of the counties in this region is higher than that of most counties in the basin, indicating that altitude exerted a certain impact on CLF. Low-low areas mean low-value and low-value agglomeration areas, which were concentrated in various cities at the junction of Anhui and Henan, counties mainly included Zhoukou, Shangqiu, Fuyang, Bozhou, and other cities, and the eastern part of some basin counties of Yancheng, Taizhou, and Nantong along the coast were also distributed. The low-high area reflects that the low value is surrounded by the high value. This type of area exhibits a small and scattered distribution, mainly scattered in a few counties in cities, such as Xuzhou and Pingdingshan. The high-low area indicates that the high value is surrounded by the low value. This type covers fewer counties, which did not appear during the study periods. Such counties are generally “outliers” and appear less frequently.

### 3.4. Influencing Factors of Cultivated Land Fragmentation

Based on Equation (11), the spatiotemporal distribution mechanism of CLF in 218 counties of the Huaihe River Basin from 2000 to 2018 was explored, through which the spatiotemporal differentiation of CLF in the Huaihe River Basin was influenced by natural factors and socioeconomic factors can be observed. Additionally, the influence of different factors on CLF varied significantly. Overall, the comprehensive influence of socioeconomic factors on CLF in the Huaihe River Basin during the study period far exceeded that of natural factors. The comprehensive influence of NDVI (X4), per capita of cultivated land area (X6), and intensity of human activities (X7) reached more than 54% of the total factor contribution rate, which was significantly stronger than other factors (Table 5). Specifically, during the study period, the influence of the altitude (X1) and slope (X2) on CLF exhibited a fluctuating upward trend, revealing that the intensity of the impact of these natural background characteristics on CLF increased. This was mainly because of the ever-growing input of agricultural science and technology, which has enabled the natural geographical advantages of a lower elevation and lower slope in the Huaihe River Basin weigh heavily in accelerating the large-scale management of cultivated land in the basin. The influence of the distance to river (X3) on CLF first increased and then decreased because with the comprehensive promotion of irrigation projects in the Huaihe River Basin, the water consumption of cultivated land was effectively guaranteed, and the dependence on important water bodies and natural precipitation weakened. NDVI (X4) exerted a high impact on CLF, but it showed a decreasing trend year by year, with a decrease of 25.6%, which is closely related to the dramatic changes of vegetation cover types and land use structure in the process of rapid urbanization. During the study period, the per capita cultivated land area (X6) and the intensity of human activities (X7) exerted a high impact on the CLF, and seemed to gain momentum. The factor contribution rate reached 0.4. On the one hand, with the improvement of the socioeconomic development level in Huaihe River Basin, the rapid population growth burdened the load of cultivated land and decreased the per capita cultivated land area (X6) continuously, leading to widespread CLF management. On the other hand, the intensifying human activities, such as disordered urban sprawl and irrational land development and utilization, badly affected the cultivated land landscape pattern.

To more clearly identify the main influencing factors of CLF and the changes in the intensity of interactions among the main influencing factors during the study period, we selected only the 10 interactions with high intensity. As shown in Table 6, the influences of each influencing factor on CLF in the Huaihe River Basin during the study period were not independent of each other, and the influence of the. interaction among factors was significantly stronger than that of a single factor, with dual-factor enhancement and nonlinear enhancement representing the majority. Specifically, in 2000, the interaction was dominated by dual-factor enhancement, and the main interaction factor intensity reached above 0.5, with the interaction intensity of human activity (X7) and NDVI (X4) the highest, reaching 0.677. In 2010, the interaction intensity of the main interaction factors increased significantly, and the dual-factor enhancement dominated. The interaction intensity between human activity intensity (X7) and per capita cultivated land area (X6) was the highest, reaching 0.729. In 2018, the dual-factor enhancement effect weakened, and the nonlinear enhancement gradually dominated. The interactive intensity of human activity (X7) and per capita cultivated land area (X6) was as high as 0.722. In general, the interaction between the intensity of human activity (X7) and other factors can better explain the CLF in the Huaihe River Basin than the interaction among other factors, indicating that the intensity of human activity (X7) played a major role in the process of CLF in the Huaihe River Basin. Mainly because the level of socioeconomic development in the Huaihe River Basin has improved, the man-land contradiction intensified, and the unreasonable human activities, such as urban sprawl and excessive reclamation, made the interaction with other factors more complex, further interfering with the landscape pattern of cultivated land.

## 4. Discussion

### 4.1. Comparing with Previous Studies

It can be observed in this study that the spatial distribution of cultivated land resources in the Huaihe River Basin was relatively uniform, the Gini coefficient of cultivated land increased year by year, and the degree of cultivated land specialization was high, which is similar to the results obtained by Liu et al. based on NDVI and the transfer matrix method [49]. Previous studies have shown that cultivated land in mountainous and hilly areas was more inclined to show the spatial separation characteristics of land structure than that in plain areas [20]. Profiting from the topographic characteristics of the wide-spread topography of the plain, the Huaihe River basin typically performs the above-mentioned cultivated land distribution characteristics. In terms of the indicators characterizing CLF, studies have constructed an evaluation index system from the aspects of the resource scale, spatial agglomeration, and convenience of utilization [1,24] but without considering the multi-collinearity of landscape indices. Principal component analysis has been applied in most relevant studies to reduce the redundancy among the selected metrics [50]. In this study, SPSS22 software was introduced for collinearity diagnostics testing of the fragmentation index selected, and the eight indicators filtered covered the three aspects of size, edge-shape, and aggregation, which can comprehensively reflect the complex process involved in fragmentation.

During the study period, the comprehensive index of CLF in the Huaihe River Basin increased by 3.8%, and the degree of fragmentation displayed an upward trend, which is consistent with the development trend of CLF across China, that is, an increasing trend of fragmentation in China was identified and southern China was characterized by more CLF than the other parts [17]. The analysis of influencing factors based on geographical detectors showed that the main factors leading to the increasing degree of fragmentation are the intensity of human activities (X7), per capita of cultivated land area (X6), and NDVI (X4), and their contribution rates were all above 0.4. Yet, based on the hierarchical linear model, Xu et al. found that the average patch area, gross domestic product (GDP), land use intensity, and urbanization rate at the county level were the main factors affecting the CLF in Jiangsu Province, and natural factors were considered to be the primary factor leading to CLF [1]. Sklenicka et al. drew the same conclusion on research concerning CLF in central and eastern Europe [25]. However, this study observed the opposite, namely the influence of socioeconomic factors on CLF weighed heavier than that of natural factors, and similar conclusions could be drawn from the study of Tan et al. [51]. This is mainly due to the rapid economic development in the Huaihe River Basin, where the impact of man-land contradiction on CLF has deepened. Meanwhile, the terrain of the basin is dominated by plain, and the impact exerted on CLF by natural factors, such as altitude and slope, was not prominent; thus, socioeconomic factors exceeded the natural ones.

Additionally, most scholars have taken socioeconomic factors [1,24], natural factors [1,20], and policy regulation factors [52] into account in the analysis touching on the. influencing factors of CLF. However, only a simple linear combination analysis of their forces weakened the chain reaction caused by the combination of multiple objectively existing factors, and lacks discussion on the intensity and mode of interaction between different influencing factors. This study found that the combination of different influencing factors of CLF produced multiple interactions, such as dual-factor enhancement and nonlinear enhancement, which increased the intensity of the impact on CLF. In addition, current research on the factors affecting CLF is mostly based on a single province or region [1,24,53], which is far-fetched to provide a reference for river basin units and important food production bases in China.

### 4.2. Policy Implications

CLF can exert a profound negative impact on the efficiency of cultivated land use, food production, and agricultural development [19,54]. In 2017, the Chinese government put forward the Rural Revitalization Strategy, which aims to accelerate the realization of agricultural and rural modernization [55]. However, the intensification of CLF has become a huge obstacle; hence, it is imperative to take measures to alleviate it. Acting as the important grain production base in China, the Huaihe River Basin covers an area of only 2.9% of China’s land area, but its cultivated land area accounts for 12% of the country’s land area [9]. However, because of the continuous urbanization occurring throughout the basin, population growth, excessive reclamation, and other high-intensity human activities in the Huaihe River Basin have resulted in frequent floods and droughts, soil erosion, ecological environment deterioration, and the associated aggravation of fragmentation, which has restricted sustainable socioeconomic development, increased the cost of food production, hindered the development of agricultural scale, and squeezed the space for future improvement of the total food production capacity in the basin, eventually seriously threatening national food security [51]. Therefore, it is of theoretical and practical significance to determine the characteristics of cultivated land resources among natural, spatial, and utilization attributes in the basin and adopt scientific and reasonable management measures to prevent and alleviate the exacerbation of the CLF phenomenon for formulating regional cultivated land utilization strategy, promoting agricultural modernization, and ensuring food security.

State-regulated consolidation is often perceived as a critical measure to tackle the CLF problem [56]. Based on the research results, the following suggestions are put forward:(1)Strictly control the occupation of cultivated land by construction land in the basin, earnestly implement the policy of requisition-compensation balance [57], strictly observe the red line of cultivated land protection, ensure the efficient utilization of cultivated land resources in the Huaihe River Basin, guarantee food security, and stabilize the position of the grain production base in the basin.(2)On the basis of protecting the ecological environment, implement differentiated conversion of cultivated land to avoid unreasonable conversion and excessive greening of cultivated land, and reduce the degree of dispersion of cultivated land patches.(3)Formulate reasonable land use planning, realize the optimization of land use structure, take the Huaihe River as the boundary, adjust paddy field and dry land differently, implement a reasonable dry land conversion project, and avoid blindly pursuing high-income land use transformation.(4)Leverage the advantages of extensive natural geography in the eastern and southern plain of the Huaihe River Basin, strengthen the support for large-scale agricultural industry management, reduce the phenomenon of land fragmentation management in the plain by support and encouragement, and improve the utilization efficiency of cultivated land resources.

### 4.3. Limitations and Future Directions

Certain limitations could be observed in this study. CLF is widely distributed globally [15,58]. CLF driven by the process of ownership fragmentation is a pervasive issue in various planning and managing activities in different countries [59]. Current research is more frequently carried out on characterizing CLF through the landscape pattern index, but the landscape scale cannot touch the cause of CLF caused by the division of ownership and the land use distribution system [60,61]. CLF is a complex process, which is formed under the combined effect of various factors, such as social economy, nature, policy, and culture. Therefore, it is not only necessary to pay attention to the impact of quantitative indicators, such as the level of socioeconomic development and natural geographical conditions on the CLF, but also to policy regulation [52], cultural customs [62], and institutional background [60,63] and other relevant indicators. Due to the limitations of data sources, it is beyond the scope of this study to indulge in a full-scale discussion on the above issues comprehensively, but these factors are crucial in exploring the study of CLF at the micro scale. Besides, the interaction between different driving force groups is complex [25], so it is extremely difficult to comprehensively study the influencing factors of CLF. Furthermore, although this study focused on the different intensities of the interactions of different influencing factors, due to the limitations of research methods, it failed to detect the driving mechanism of the interaction of different factors, which needs further improvement in follow-up study. Simultaneously, based on the understanding of the CLF in the county unit of the Huaihe River Basin, we will further explore it in township units and even villages in the future, with a view to providing support for the governance of CLF in the basin and ensuring food security.

## 5. Conclusions

Based on the land use remote sensing data from 2000 to 2018, Fragstats4.2 (Oregon State University, Corvallis, OR, USA) and ArcGIS10.3 were adopted, the spatiotemporal distribution characteristics of CLF index at county level in the Huaihe River Basin were measured by the entropy weight method, and multiple influencing factors of CLF were explored with the help of the geographical detector. The main conclusions are as follows:(1)From 2000 to 2018, the Gini coefficients of cultivated land were 0.105, 0.108, and 0.113, respectively, increasing year by year but less than 0.15. More than 56% of the counties in the basin showed a cultivated land location entropy greater than 1, and the degree of specialization of cultivated land continued to increase. Whereas the location entropy of counties in Zhengzhou, Xinyang, Huaibei, and other cities showed a downward trend.(2)From 2000 to 2018, PLAND, AREA_AM, COHESION, and AI, which are four negative indicators of cultivated land fragmentation in the Huaihe River Basin, showed a decreasing trend year by year while the positive indicators of PD and SPLIT increased year by year. Overall, the spatial and temporal differences of CLF in the Huaihe River Basin were distinguished, and the degree of CLF increased, but the degree of CLF in some areas showed a coexistence of an increase and decrease.(3)The county distribution pattern of the CLF degree with random and agglomeration was generally stable. High-high areas were concentrated in some counties in Linyi, Xuzhou, Zhengzhou, and other cities; low-low areas were concentrated in cities at the junction of Anhui and Henan, with Zhoukou, Shangqiu, Fuyang, and Bozhou the main counties. The low-high areas had a small and scattered distribution, mainly distributed in a small number of counties in Xuzhou, Pingdingshan, Linyi, and other cities. High-low areas did not appear in the three time breakpoints, indicating that within the basin, there was no significant difference in the degree of CLF between adjacent counties, and no obvious polarization phenomenon was observed.(4)During the study period, the spatiotemporal differentiation of CLF in the Huaihe River Basin was affected by multiple factors, such as nature, socioeconomic, etc., and the influence of different factors on CLF was significantly different. The comprehensive influence of socioeconomic factors was significantly stronger than that of natural factors. The influence of NDVI (X4), per capita of cultivated land area (X6), and intensity of human activity (X7) was significantly stronger than that of other factors, with the factor contribution rate above 0.4. The intensity of human activity (X7) played a major role in the process of CLF. The influence of various factors on the CLF in the Huaihe River Basin was not independent of each other. The interaction effect among the factors was stronger than that of a single factor, with dual-factor enhancement dominant and nonlinear enhancement the supplement.

We hope that the results of this study and the proposed policy recommendations can provide references for identifying and alleviating CLF in important grain production areas to ensure regional food security.

## Figures and Tables

**Figure 1 ijerph-19-00138-f001:**
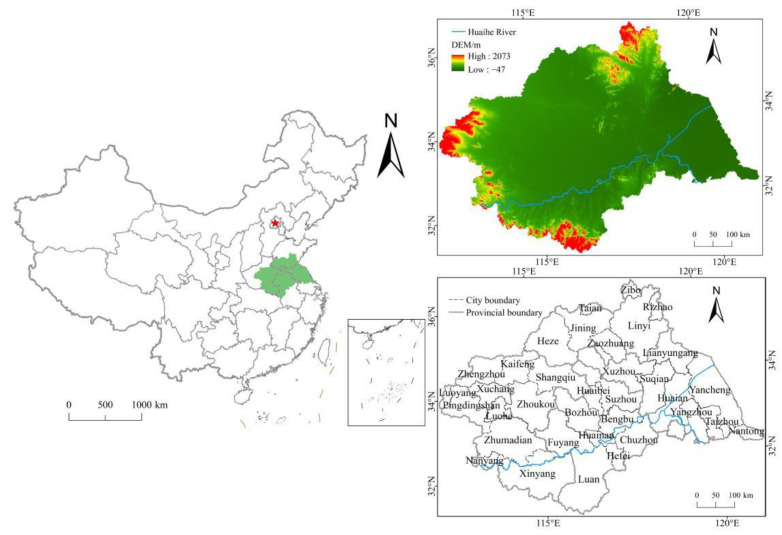
Location of the study area.

**Figure 2 ijerph-19-00138-f002:**
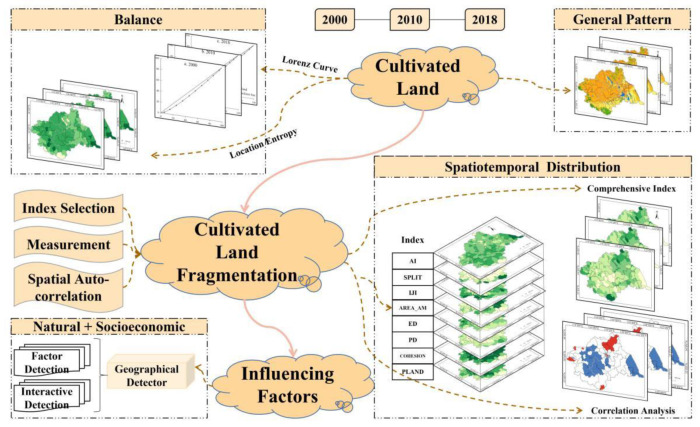
Conceptual framework for measuring cultivated land fragmentation and detecting its influencing factors.

**Figure 3 ijerph-19-00138-f003:**
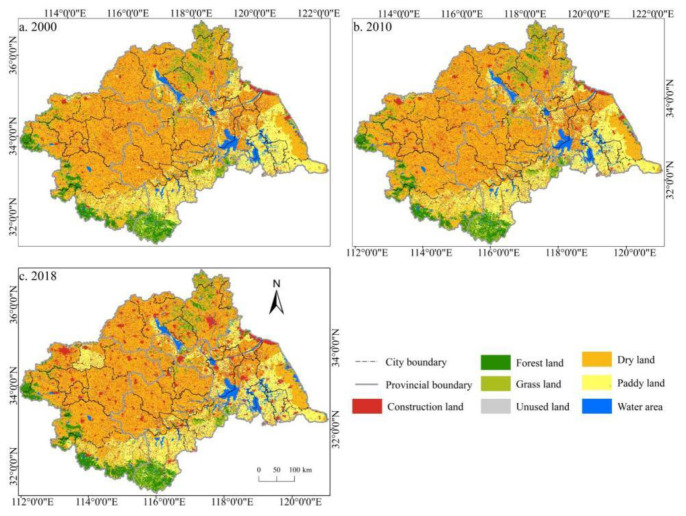
Spatial distribution of land use in the Huaihe River Basin from 2000 to 2018. Notes: (**a**) the spatial distribution of land use in 2000; (**b**) the spatial distribution of land use in 2010; (**c**) the spatial distribution of land use in 2018.

**Figure 4 ijerph-19-00138-f004:**
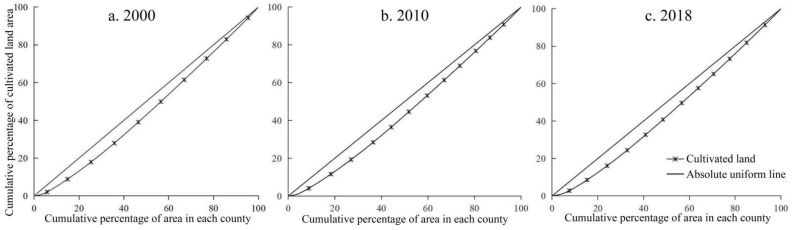
The Lorentz curve of cultivated land in the Huaihe River Basin from 2000 to 2018. Notes: (**a**) the Lorentz curve of cultivated land in 2000; (**b**) the Lorentz curve of cultivated land in 2010; (**c**) the Lorentz curve of cultivated land in 2018.

**Figure 5 ijerph-19-00138-f005:**
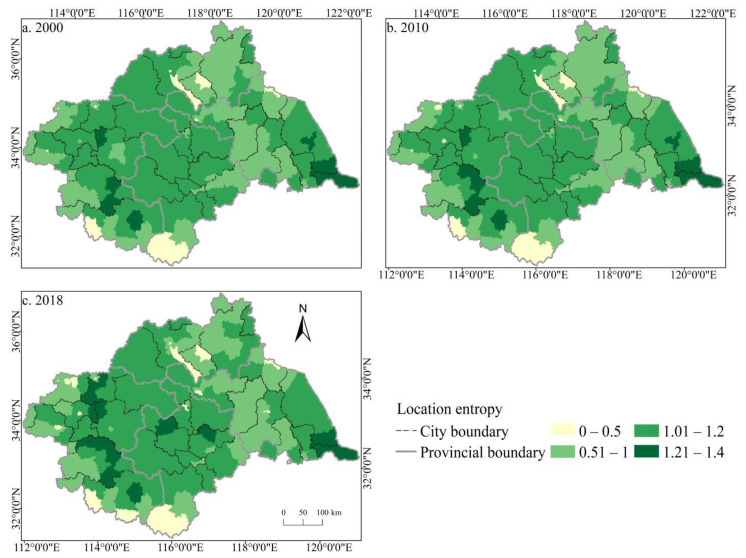
Spatial distribution of the cultivated land location entropy in the Huaihe River Basin from 2000 to 2018. Notes: (**a**) the spatial distribution of cultivated land location entropy in 2000; (**b**) the spatial distribution of cultivated land location entropy in 2010; (**c**) the spatial distribution of cultivated land location entropy in 2018.

**Figure 6 ijerph-19-00138-f006:**
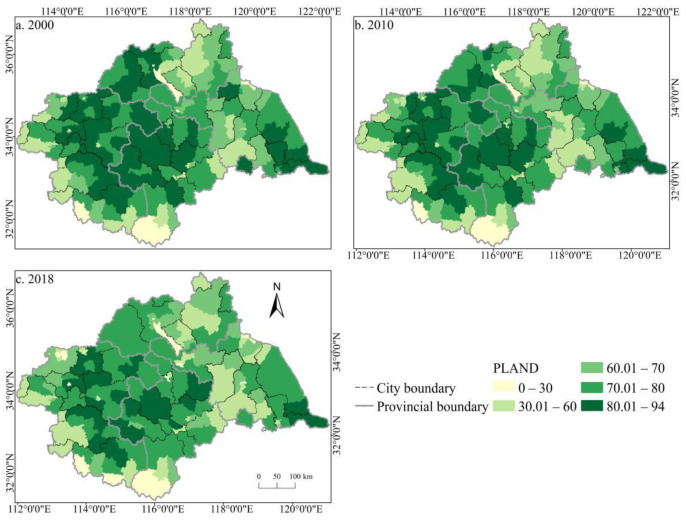
Spatial distribution of PLAND of cultivated land in the Huaihe River Basin from 2000 to 2018. Notes: (**a**) the spatial distribution of PLAND of cultivated land in 2000; (**b**) the spatial distribution of PLAND of cultivated land in 2010; (**c**) the spatial distribution of PLAND of cultivated land in 2018. PLAND: Percentage of landscape.

**Figure 7 ijerph-19-00138-f007:**
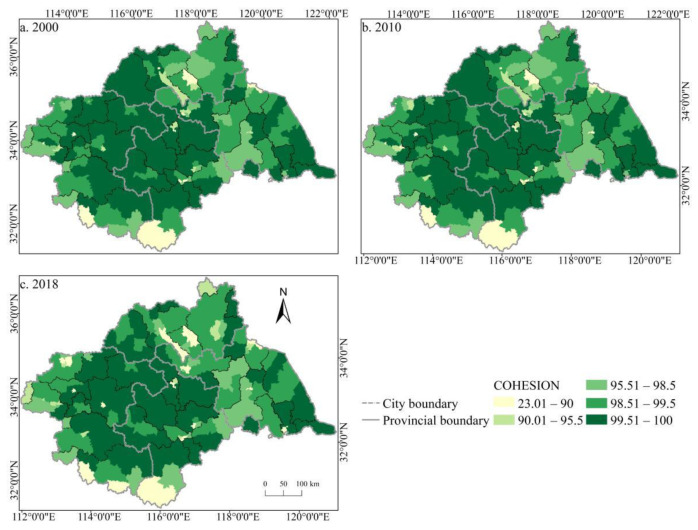
Spatial distribution of COHESION of cultivated land in the Huaihe River Basin from 2000 to 2018. Notes: (**a**) the spatial distribution of COHESION of cultivated land in 2000; (**b**) the spatial distribution of COHESION of cultivated land in 2010; (**c**) the spatial distribution of COHESION of cultivated land in 2018. COHESION: Patch cohesion index.

**Figure 8 ijerph-19-00138-f008:**
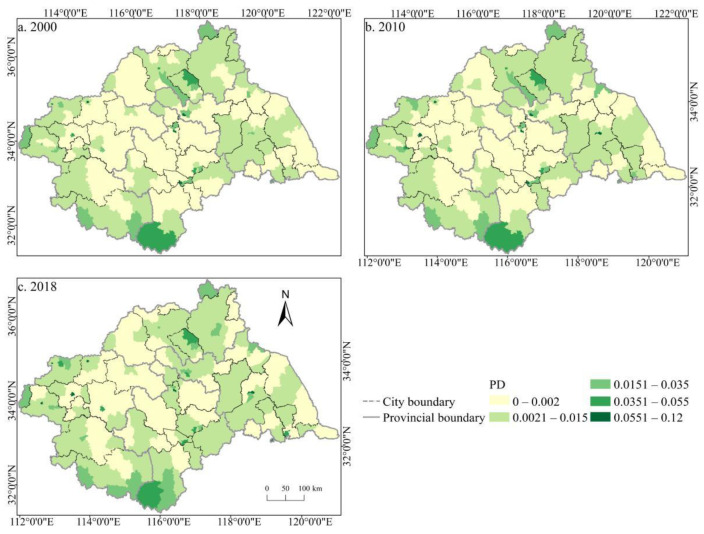
Spatial distribution of PD of cultivated land in the Huaihe River Basin from 2000 to 2018. Notes: (**a**) the spatial distribution of PD of cultivated land in 2000; (**b**) the spatial distribution of PD of cultivated land in 2010; (**c**) the spatial distribution of PD of cultivated land in 2018. PD: Patch density.

**Figure 9 ijerph-19-00138-f009:**
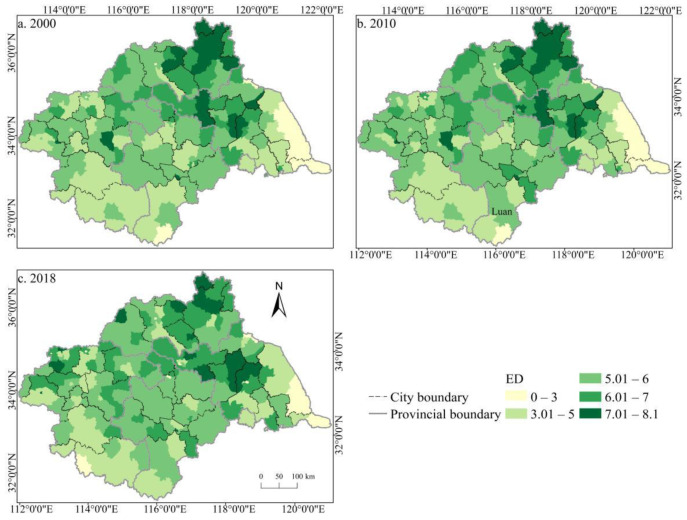
Spatial distribution of ED of cultivated land in the Huaihe River Basin from 2000 to 2018. Notes: (**a**) the spatial distribution of ED of cultivated land in 2000; (**b**) the spatial distribution of ED of cultivated land in 2010; (**c**) the spatial distribution of ED of cultivated land in 2018. ED: Edge density.

**Figure 10 ijerph-19-00138-f010:**
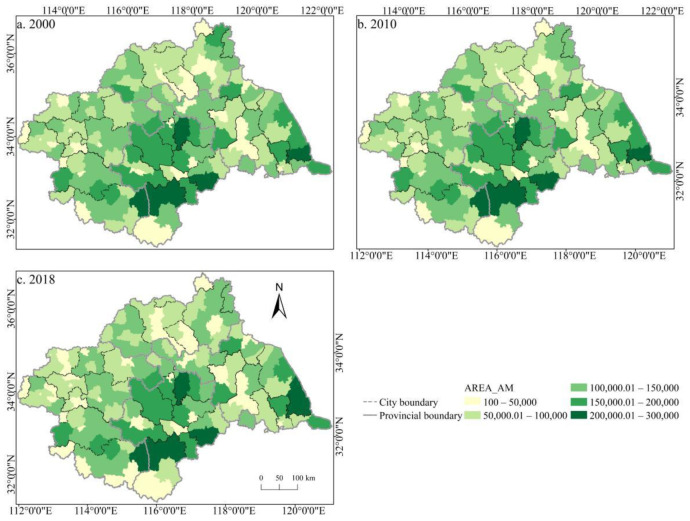
Spatial distribution of AREA_AM of cultivated land in the Huaihe River Basin from 2000 to 2018. Notes: (**a**) the spatial distribution of AREA_AM of cultivated land in 2000; (**b**) the spatial distribution of AREA_AM of cultivated land in 2010; (**c**) the spatial distribution of AREA_AM of cultivated land in 2018. AREA_AM: Area-weighted mean patch area.

**Figure 11 ijerph-19-00138-f011:**
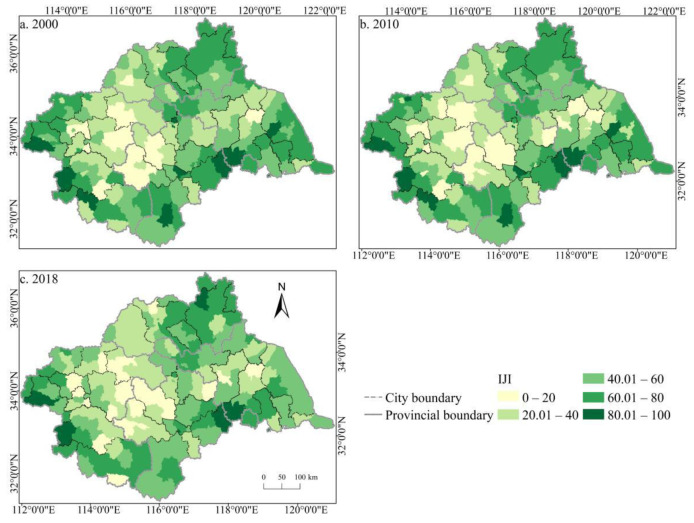
Spatial distribution of IJI of cultivated land in the Huaihe River Basin from 2000 to 2018. Notes: (**a**) the spatial distribution of IJI of cultivated land in 2000; (**b**) the spatial distribution of IJI of cultivated land in 2010; (**c**) the spatial distribution of IJI of cultivated land in 2018. IJI: Interspersion and juxtaposition index.

**Figure 12 ijerph-19-00138-f012:**
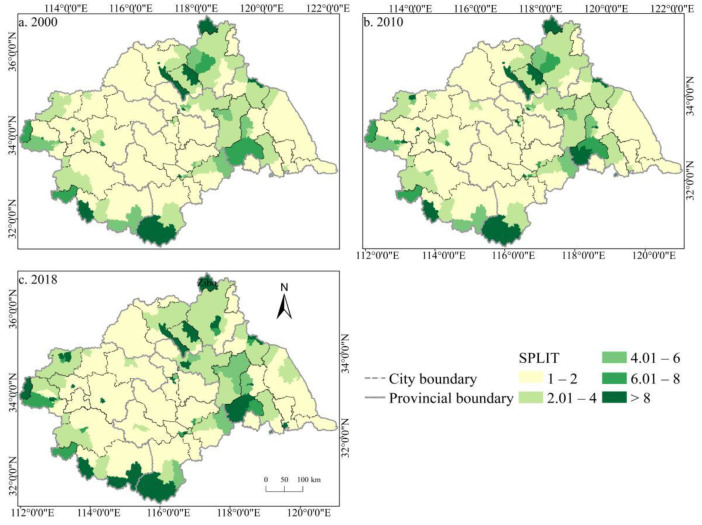
Spatial distribution of SPLIT of cultivated land in the Huaihe River Basin from 2000 to 2018. Notes: (**a**) the spatial distribution of SPLIT of cultivated land in 2000; (**b**) the spatial distribution of SPLIT of cultivated land in 2010; (**c**) the spatial distribution of SPLIT of cultivated land in 2018. SPLIT: Splitting index.

**Figure 13 ijerph-19-00138-f013:**
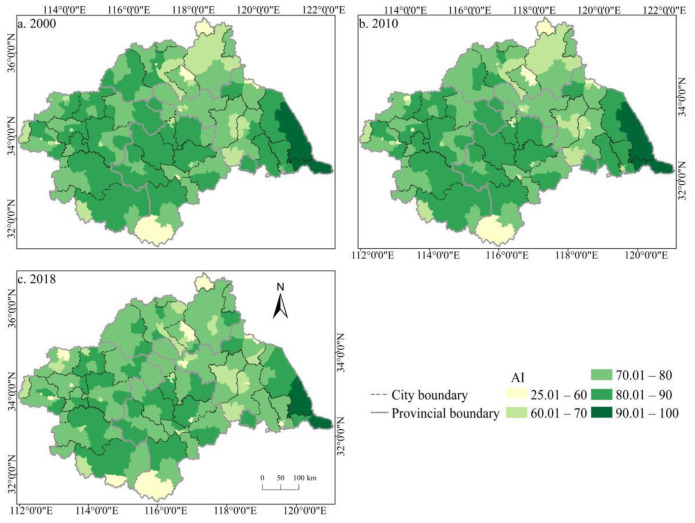
Spatial distribution of AI of cultivated land in the Huaihe River Basin from 2000 to 2018. Notes: (**a**) the spatial distribution of AI of cultivated land in 2000; (**b**) the spatial distribution of AI of cultivated land in 2010; (**c**) the spatial distribution of AI of cultivated land in 2018. AI: Aggregation index.

**Figure 14 ijerph-19-00138-f014:**
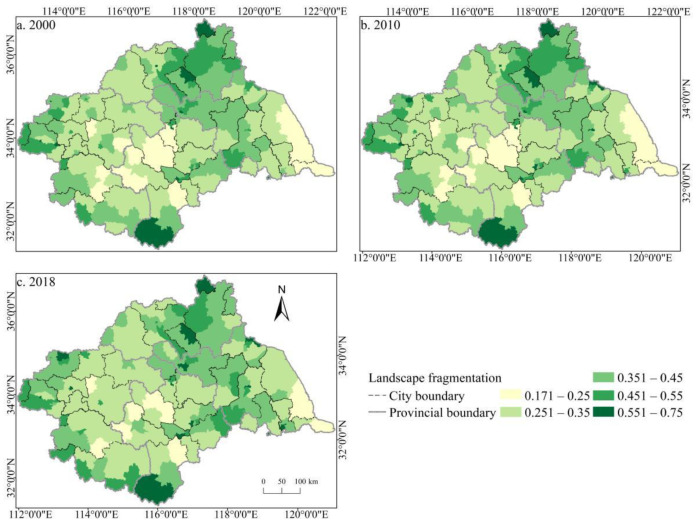
Spatial distribution of the comprehensive fragmentation degree of cultivated land in the Huaihe River Basin from 2000 to 2018. Notes: (**a**) the spatial distribution of comprehensive fragmentation degree of cultivated land in 2000; (**b**) the spatial distribution of comprehensive fragmentation degree of cultivated land in 2010; (**c**) the spatial distribution of comprehensive fragmentation degree of cultivated land in 2018.

**Figure 15 ijerph-19-00138-f015:**
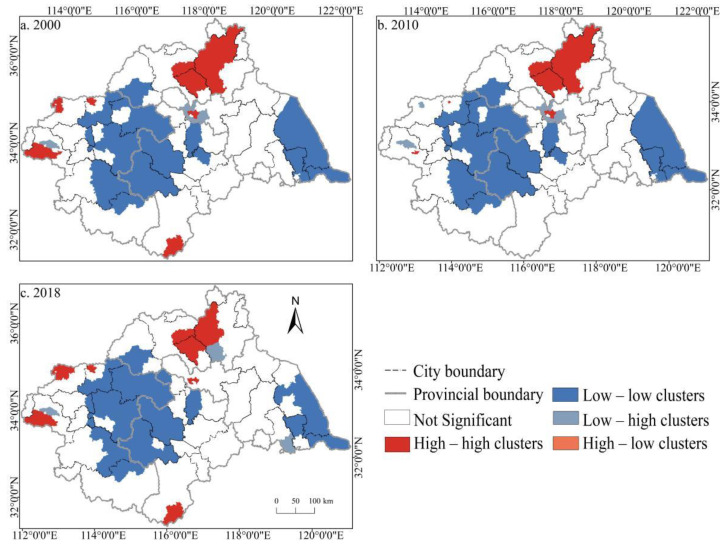
Local indicators of the spatial association (LISA) map of cultivated land fragmentation in the Huaihe River Basin from 2000 to 2018. Notes: (**a**) LISA map of cultivated land fragmentation in 2000; (**b**) LISA map of cultivated land fragmentation in 2010; (**c**) LISA map of cultivated land fragmentation in 2018.

**Table 1 ijerph-19-00138-t001:** Multi-collinearity diagnosis table of landscape metrics.

Metrics	PLAND	PD	ED	AREA_AM	SHAPE_AM
VIF	7.473	4.433	5.750	7.131	18.309
Metrics	FRAC_AM	IJI	COHESION	SPLIT	AI
VIF	11.263	1.223	5.409	1.823	9.625

**Table 2 ijerph-19-00138-t002:** Cultivated land fragmentation index and its description.

Aspects	Landscape Pattern Index	Calculation Formula	Description of Index
Size	Percentage of landscape (PLAND)	$PLAND=\frac{\sum_{j=1}^{n}a_{ij}}{LA}$ where *a_ij_* is the area of patch *ij*; *LA* is the total landscape area of the study area.	PLAND indicates the area of a certain patch type accounts for the percentage of the total landscape area, and the value tends to be 0. The scarcer the patch type, the more fragmented the landscape pattern.
	Patch density (PD)	$PD=\overset{m}{\underset{i=1}{\sum }}N_{i}/LA$ where *m* is the total number of landscape types; *LA* is the total landscape area of the study area; *N_i_* is the patch number of landscape i.	PD indicates the degree of influence of the patch boundary of the landscape type on the entire landscape. The larger the value, the more concentrated the patch type is distributed in the landscape.
Edge—shape	Edge density (ED)	$ED=\overset{m}{\underset{i=1}{\sum }}\overset{m}{\underset{j=1}{\sum }}P_{ij}/LA$ where *P_ij_* is the boundary length between type *i* and type *j* landscape element patches.	ED refers to the degree of landscape type segmentation by element boundary, which is a direct reflection of landscape fragmentation. The larger the value, the more fragmented the landscape pattern of this type of element.
	Area-weighted mean patch area (AREA_AM)	MPS=LA/NP where *LA* is the total landscape area of the study area; *NP* is the number of patches in the landscape.	AREA_AM reflects the degree of fragmentation of a certain type of landscape in landscape structure analysis. The higher the value, the lower the degree of fragmentation.
Aggregation	Interspersion and juxtaposition index (IJI)	$IJI=\frac{-\sum_{i=1}^{m}\sum_{j=i+1}^{m}\left[\left(\frac{E_{ij}}{E}\right)ln\left(\frac{E_{ij}}{E}\right)\right]}{ln\left[0.5m\left(m-1\right)\right]}\times 100$ where *E_ij_* is the adjacent edge length between the type *i* and the type *j* of feature patch; *m* is the total number of landscape types; *E* is the edge length of the whole landscape.	IJI refers to the adjacent probability between certain patch element and other patches. The higher the value, the more adjacent other patch types, and the more fragmented the landscape of this type.
	Patch cohesion index (COHESION)	$COHESION=\left[1-\frac{\sum_{i=1}^{m}\sum_{j=1}^{n}P_{ij}}{\sum_{i=1}^{m}\sum_{j=1}^{n}P_{ij}\sqrt{a_{ij}}}\right]\times {\left[1-\frac{1}{\sqrt{Z}}\right]}^{-1}\times 100$ where *P_ij_* represents the perimeter of patch *i* of landscape type *j*; *a_ij_* represents the area of patch *i* of landscape type *j*; *Z* represents the number of patches in the landscape.	COHESION indicates the degree of agglomeration between different types of patches, the larger the value, the higher the degree of combination between dominant types of patches, and the lower the degree of fragmentation of this type of landscape.
	Splitting index (SPLIT)	$SPLIT=\frac{LA^{2}}{\sum_{i=1}^{m}\sum_{j=1}^{n}a_{ij}^{2}}$ where *LA* is the total landscape area of the study area; *a_ij_* is the area of patch *j* of landscape type *i*; *n* is the number of patches of landscape type.	SPLIT refers to the degree of separation of landscape element. The larger the value is, the more dispersed among the same patch types and the higher the degree of fragmentation.
	Aggregation index (AI)	$AI=\left[\frac{g_{ij}}{max\to g_{ij}}\right]$ where *g_ij_* is the number of similar adjacent patches of the landscape patch type	AI refers to the degree of agglomeration between patches of a certain type of landscape element. The larger the value, the more agglomerated patches of this type of element and the lower the degree of fragmentation.

**Table 3 ijerph-19-00138-t003:** Description of the driving factors classification of the geographic detector.

Driving Factors	Classification Method	Level	Level Description
Altitude	NaturalBreaks	1–6	Calculation with Arc Toolbox/Spatial Analyst Tools/Reclass of ArcGIS10.3
Slope	NaturalBreaks	1–6	1. 0~5; 2. 6~10; 3. 11~15; 4. 16~20; 5. 21~25; 6. >25
Distance to river	NaturalBreaks	1–5	Calculation with Arc Toolbox/Spatial Analyst Tools/Reclass of ArcGIS10.3
NDVI	Zhang et al. [46]	1–5	1. ≤0.2; 2. 0.2~0.4; 3. 0.4~0.6; 4. 0.6~0.8; 5. 0.8~1
Average annual precipitation	NaturalBreaks	1–5	Calculation with Arc Toolbox/Spatial Analyst Tools/Reclass of ArcGIS10.3
Per capita cultivated land area	NaturalBreaks	1–5	Calculation with Arc Toolbox/Spatial Analyst Tools/Reclass of ArcGIS10.3
Intensity of human activities	Li et al. [47]	0–10	0. sparse woodland, shrub, sparse grass, barren land; 1. river, reservoir, ponds, tidal flat, natural and plantation forest land, Moderate grass; 2. other woodland, dense grass; 7. cultivated land; 8. rural settlements; 9. industrial land; 10. Urban built-up
Population density	Ge et al. [48]	1–5	1. 0~60; 2. 61~150; 3. 151~300; 4. 301~500; 5. >500
Distance from road network	NaturalBreaks	1–5	Calculation with Arc Toolbox/Spatial Analyst Tools/Reclass of ArcGIS10.3

**Table 4 ijerph-19-00138-t004:** Multi-collinearity diagnosis table of landscape metrics.

Landscape Index	PLAND (#/100 ha)	PD (m/km^2^)	ED	AREA_AM (ha)	IJI (%)	COHESION	SPLIT	AI (%)
2000	69.236	0.009	5.350	83,991.047	49.444	97.346	10.220	76.025
2010	67.495	0.010	5.410	82,716.491	48.591	96.792	12.562	75.257
2018	63.603	0.011	5.357	79,706.491	46.155	93.374	17.511	73.439

**Table 5 ijerph-19-00138-t005:** Contribution rate of impact factors from 2000 to 2018.

Year	X1	X2	X3	X4	X5	X6	X7	X8	X9
2000	0.182	0.163	0.147	0.512	0.063	0.384	0.376	0.150	0.151
2010	0.169	0.159	0.165	0.406	0.096	0.424	0.378	0.259	0.141
2018	0.186	0.169	0.141	0.381	0.143	0.408	0.433	0.237	0.144

Notes: X1: altitude; X2: slope; X3: distance to river; X4: normalized difference vegetation index (NDVI); X5: average annual precipitation; X6: per capita cultivated land area; X7: intensity of human activities; X8: population density; X9: distance from road network.

**Table 6 ijerph-19-00138-t006:** Main interaction factors and changes.

2000		2010		2018	
Interaction Factors	Interaction Intensity	Interaction Factors	Interaction Intensity	Interaction Factors	Interaction Intensity
X3∩X4	0.555 *	X1∩X4	0.532 *	X5∩X4	0.544 ^#^
X5∩X4	0.567 *	X8∩X4	0.570 *	X5∩X7	0.558 ^#^
X8∩X4	0.585 *	X6∩X4	0.580 *	X8∩X7	0.571 *
X8∩X6	0.588 ^#^	X5∩X4	0.582 ^#^	X8∩X6	0.579 *
X1∩X6	0.590 ^#^	X8∩X7	0.594 *	X1∩X4	0.604 ^#^
X1∩X4	0.594 *	X7∩X4	0.602 *	X1∩X6	0.608 ^#^
X6∩X4	0.596 *	X2∩X6	0.609 ^#^	X2∩X4	0.609 ^#^
X2∩X6	0.598 ^#^	X8∩X6	0.618 *	X2∩X6	0.612 ^#^
X2∩X4	0.619 *	X1∩X6	0.627 ^#^	X7∩X4	0.674 *
X7∩X4	0.677 *	X7∩X6	0.729 *	X7∩X6	0.722 *

Notes: * means dual-factor enhancement; # means nonlinear enhancement. X1: altitude; X2: slope; X3: distance to river; X4: normalized difference vegetation index (NDVI); X5: average annual precipitation; X6: per capita cultivated land area; X7: intensity of human activities; X8: population density.

## Data Availability

The data that support the findings of this study are available from the corresponding author upon reasonable request.

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
