# Peer review of "Cultivated Land Fragmentation and Its Influencing Factors Detection: A Case Study in Huaihe River Basin, China"

_ijerph, 2021, doi:10.3390/ijerph19010138_

Round 1

Reviewer 1 Report

General Comments:

An interesting manuscript investigating cultivated land fragmentation in Huaihe River Basin and interactive factors that that illustrate the progression of the fragmentation. I have a few comments, mainly about clarifying certain aspects of the manuscript.

Specific Comments:

  1. Lines 27-38. It is unconventional to have a list of the results in the abstract. I appreciate that it is reflective of the conclusion, but consider removing the list element.
  2. Line 49. Ecology is not a function, consider reformatting.
  3. Line 53. I do not understand what is meant with ‘forced the transformation of land use patterns’. Reformat it to make it easier to understand.
  4. Line 54. Cultivated land fragmentation is extensively talked about but not defined. Include a clear definition that specifies what the concept means.
  5. Lines 89-102. This section describing previous research is repetitive and not informative. Synthesize these lines to describe the current state of the research, what is generally known about the CLF rather than listing each study.
  6. Lines 486. Use the full name of the LISA map, not just the acronym.
  7. Throughout the manuscript there are mentions of ‘counties (districts)’. Specify in the methods one of these concepts (they are basically identical) and use one of them instead of both.

Figures and Tables:

  1. Figures 3 & 5-13. Needs to be bigger. It is almost impossible to read the figure legends and other figure elements. Increase the size of the figures in general and remove the city/province names so that the data itself is visible.
  2. Figure 3. The figure is misleading, it suggests that 100% of the area are cultivated land to some extent, which is doubtful. There must be other areas that are not cultivated, for example urban areas. Include these types of areas in the figure and legend as well for clarity.
  3. Figure 4. Need axes.
  4. Figures 6-13. It would be helpful for the reader if a short sentence or description of the variable being displayed here. I know the variables are described in earlier tables but perhaps a short description would help the reader, since it is difficult to remember exactly what all the variables are by just their initials and acronyms.
  5. Table 5. The table description needs extending and the names of the variables needs to be there instead of X1, X2 etc. It is impossible to tell what the table represents without reading the entire text, the table needs to be readable as stand-alone.
  6. Table 6. Same here as with Table 5. It also needs to clarified either in the text or the table description why only these sets of interactions were included in the table, and not all. Perhaps all interactions could be put into supplementary material, if the reader is interested in a specific interaction that is not part of the main manuscript.

Reviewer 2 Report

The methods and results of this paper are very thoroughly and clearly described, with ample detail, making it easy to follow exactly what was done and what was found. The rationale (i.e. framing) and implications of the results are relatively thin, though. My comments pertain to these.

  1. Why is fragmentation a problem in terms of cultivated land? I don't think that this basic point was adequately established in the introduction. Loss of farmland could lead to reduced productivity (but not always, more on that below), but what is the problem with fragmentation per se? In terms of natural habitats (e.g. forests, wetlands), fragmentation can lead to declining populations of certain species. Cultivated lands provide habitat, which could be part of the reason, but is it more than land? Do fragmented farmlands present operational or logistical hurdles? Does per-unit productivity suffer? In other words, you need to disentangle loss of cultivated land from fragmentation of cultivated land and make a case for each?
  2. Where total acreage of cultivated land decreases, can you identify what land covers replace it? Using supporting literature, you seem to suggest that usually it is urban land - but is that what the data show in your case? If cultivated land is being replaced with forests/wetlands/grasslands, this decline may be associated with net positive  ecosystem services.
  3. Do you have access to data or reports that would allow you to say whether productivity (i.e. food production) in this region has declined or increased over the study period? Without this, it is hard to interpret the implications for fragmentation.

These three points are valuable (essential, actually) in order to be able to interpret your results in a meaningful way. You demonstrate very thoroughly that fragmentation is happening, but without knowing the answers to the three points above I don't know whether that is good or bad. For example, on one hand, CLF could be the result of increasing urban/suburban land in the face of flat or even declining productivity gains, resulting in loss of ecosystem services as well as agricultural productivity, increases in food insecurity, pollution, habitat destruction, etc. On the other hand, the story could very different: CLF could be resulting from agricultural abandonment in the face of productivity gains, with returns of cultivated lands to forests and shrublands - leading to lower pollution, more habitat, and no increase of food security. Your language implies the first case, but the second case is also very possible. In my part of the world, the northeastern U.S. - CLF was driven by the latter scenario and most people consider it to have been a net positive transition.

Reviewer 3 Report

In my opinion it is a very well prepared manuscript in terms of technical and content. Well-chosen research methods, appropriate research area, high scientific reliability of the Authors.

I think that the presented topic is original and brings new insights into cultivated land fragmentation and its influencing factors detection.

The research results presented in the manuscript confirm and continue the results and studies presented in other materials and publications. The authors presented the research carried out in Huaihe River Basin in China in an interesting way. This brings additional knowledge on this topic.

The methodology, course of work, their analysis, and interpretation have been properly planned, carried out and described in the text.

The results were correctly presented and relate to the issues discussed in the manuscript.

All references cited have been properly prepared and used. The introduction to the topic has been properly and meticulously prepared.

My only remark is to the figures - there are too many of them and they are too small and not very legible - especially maps (Fig, 3,5-17). Figure 2 is an important aspect from a methodological point of view - it should be larger and of better quality. - This is not a note to the manuscript, but a suggestion.

A very well prepared mauscript!
